# Influence of Parenting Guilt on the Mental Health among Single-Parent Women: Multiple Additive Moderating Effect of Economic Well-Being and Level of Education

**DOI:** 10.3390/healthcare11131814

**Published:** 2023-06-21

**Authors:** Anna Kim, Sesong Jeon, Inah Park

**Affiliations:** 1Welfare and Family Policy Research Team, Women’s Family Policy Headquarters, Ulsan Public Agency for Welfare Family Promotion Social Service, Ulsan 44000, Republic of Korea; skyplane1022@uwfdi.re.kr; 2Major in Child & Family Studies, School of Child Studies, College of Human Ecology, Kyungpook National University, Daegu 41566, Republic of Korea; inahda@knu.ac.kr

**Keywords:** mental health, parenting guilt, economic well-being, level of education, single-parent women, multiple additive moderating effect

## Abstract

This study explored how the core problems (e.g., parenting, economy, and education level) of single-parent women affect their mental health. Although parenting guilt, economic well-being, and level of education are important variables that affect the mental health of single parents, there is no study that examines the interaction effect between them together. Therefore, this study examined the moderating effects of economic well-being and level of education on the relationship between parenting guilt and mental health in 419 single-parent women. In addition, it was verified whether there was a multiple additive modulation effect when they were put in at the same time. As a result, the higher the parenting guilt and the lower the economic well-being, the higher the level of mental health pain, but the level of education had no statistical significance. However, the interaction term between parenting guilt and education level had statistical significance, while the interaction term between parenting guilt and economic well-being did not produce significant results. These findings suggest the importance of education for single-parent women’s mental health as well as the need to establish policies that allow them to have sufficient time and room for child rearing.

## 1. Introduction

According to the United States Census Bureau in 2022, nearly 80% of the approximately 11 million single-parent families with children under the age of 18 consist of single mothers [1]. Considering the number of young single parents who have not been reported in the statistics office data due to social stigma, the actual number of single-parent families with single-parent mothers is expected to be much higher. The economic burden, difficulties in raising children, and social stigma that arise for single parents cause loss, inferiority, anger, depression, and social contraction in single parents [2]. In particular, bereavement, separation, and divorce from spouses are some of the major family stress events for individuals [3]. According to many previous studies, parents from single-parent families have higher levels of emotional pain, such as stress, depression, anxiety, alcohol use disorders, and somatization disorder, than parents from two-parent families [4,5,6,7]. Researchers found that single mothers are twice as likely to have symptoms of depression or anxiety as married women with partners [5,8]. Parents’ depression or anxiety leads to parenting stress, resulting in negative parenting behaviors such as coercive or neglectful treatment of their children [9], causing internalization symptoms and externalization problem behaviors such as depression or anxiety in their children [7]. Given that depression reduces motivation for life and increases the risk of extreme suicide, strategies to support and prepare single parents for a healthy family life are required. 

The psychological and emotional difficulties faced by female householders in single-parent families are commonly related to child rearing and economic difficulties [10]. Parenting guilt is defined as the psychological guilt and shame that parents feel during the parenting process [11]. In the context of work-family conflicts, research has focused primarily on working mothers’ maternal guilt for surrogate parenting and day care when they are not physically present with their children [12,13]. Parents from single-parent families feel sorry for their children for being raised in a single-parent family compared to a two-parent family. Factors such as economic burden, work-family conflict, and social stigma make parents of single-parent families feel guilty about parenting. In addition, elevated parenting guilt functions as a negative factor for single parents’ parenting conduct, leading to indifference, neglect, and abuse, which have detrimental effects on the cognitive, psychological, and behavioral characteristics of children [14]. The guilt of being a parent was very strong when a mother had to balance her economic responsibilities with raising her children and when her performance was less than expected [15]. Parenting guilt was the strongest predictor of a mother’s subjective well-being and was a stronger predictor of depression than spouse and friend relationship factors [16]. On the other hand, Lee [17] also suggested that parenting guilt not only causes negative effects on mental health but also looks at positive aspects such as the hope and opportunity for growth that mothers want in contradiction with the current situation. Therefore, considering that guilt is relatively easier for women to feel than it is for men and that the majority of single parents are mothers, it is necessary to examine how parenting guilt can affect the mental health of single-parent women.

Next, this study aims to examine how single-parent women’s economic well-being is perceived and how it is related to mental health. Economic well-being, which means subjective satisfaction with the objective household economy, consists of income adequacy, income security, and income stability [18,19]. Previous studies have shown consistently that single-parent families are more likely to experience poverty than two-parent families [20,21,22]. In particular, female heads of households are experiencing economic poverty due to lower socioeconomic conditions than male heads of households [23]. The lower the household income, the greater the importance of individual subjective welfare. According to the results of the “Single-parent Family Survey” recently conducted in Korea, the average monthly income of single-parent families is 56.5% of the total household income, which is half the level. Although 84% of single parents are employed, they are concentrated among non-regular workers and low positions, and as a typical working force, the income level of female heads of households was lower than that of male heads of households. The results of more than 80% of single parents who said childcare and education expenses were the most urgent supported their low economic satisfaction and stability [24]. Economic poverty has created low self-esteem for single parents and led them to experience anxiety about the future [25,26].

In addition, single-parent women are likely to be in situations of conflict where they have to choose between economic rights and custody. In order to deal with the problem of balancing work and family life, poor single mothers put raising their children first [27,28,29] or chose to work instead of raising their children [30]. As a moderating variable, the sense of economic well-being will allow us to identify the structural vulnerability of single parents, who have to bear the economic burden and handle parenting. 

At the same time, the level of education of single parents is a major factor in how well they can support themselves financially [22]. Higher education helps single mothers get out of poverty and earn more money, which makes them less financially vulnerable [31]. It is also an opportunity to complete studies with a desire for education, increase family stability through employment and economic activities, and restore self-realization and self-esteem by setting an example for children. On the other hand, the low level of education of single parents acts as a vulnerable factor in employment opportunities, so they do not even get opportunities for economic activities, and at low wages, life satisfaction decreases, weakening self-esteem [31]. It can be inferred that the level of education of a single parent will have a moderating function according to the following two hypotheses: One is the ‘compensation hypothesis’, and the other is the ‘floor effect hypothesis’. The compensation hypothesis is the view that the mother’s high level of education plays a compensatory role in offsetting the negative consequences of single parenthood on their children. Single-parent women with a higher level of education are more financially stable and have better mental health than single-parent women with a lower level of education (for example, they know about the negative effects of single-parent families and how to handle being a good parent) [32]. In a following study, mothers with low education levels also saw their children as less vulnerable to the negative effects of family structure through the “floor effect hypothesis”, which could not be worse than now [32,33].

When taken together, this study seeks to determine whether economic well-being and level of education moderate the influence of parenting guilt on mental health. A single parent’s perception of economic well-being is correlated with their parental guilt. Women who are single parents are employed in transitory part-time or non-regular contract positions and remain in low-income groups. In actuality, they feel economic helplessness and depression because their children lack education or opportunity. However, a single parent’s economic well-being can serve as a stress-coping mechanism that improves their mental health. Since economic well-being depends more on individual satisfaction with resources than on the quantity of resources they possess, the lower the household income, the greater its significance. In addition, the education level of single-parent women can moderate the relationship between parenting guilt and mental health. This is due to the fact that the more education single-parent women have, the more resources they have to deal with quality parenting in advance and the more psychological fortitude they have. Although parenting guilt, economic well-being, and level of education are significant variables that affect the mental health of single parents, it is difficult to locate research that examines their interactive effects. Accordingly, in this study, the following research problems were specifically set: 

Question 1. Does parenting guilt, economic well-being, and education level affect the mental health of single-parent women?

Question 2. Does the economic well-being and education level of single-parent women moderate the effect of parenting guilt on mental health?

## 2. Methods

### 2.1. Sampling

This study conducted online survey of single-parent families residing in Ulsan, Korea, from 30 July 25 to August 2021 with the assistance of the person in charge of the single-parent family support project. Out of the 500 single-parent families that were used as a sample, this study only looked at 419 single-parent women. Their demographic information is shown in Table 1. Half of the study subjects were in their 40s (54.9%), had graduated from high school (51.1%), and had an income level of more than 1 million won to less than 2 million won (57.8%). In addition, more than 5 years to less than 10 years (33.7%) and less than 3 years (29.1%) were the most common cases of single parents. Most of the family members were less than three (67.5%), and the areas living in Ulsan were relatively evenly distributed. 

### 2.2. Analytic Plan

This study was analyzed using the SPSS 25.0 (IBM Co., Armonk, NY, USA) and PROCESS Macro Version 4.1 programs [34]. First, frequency analysis and descriptive statistical analysis (mean, standard deviation, percentage, etc.) were conducted to understand the demographic characteristics of the participants and the characteristics of the major variables. Second, Cronbach’s α coefficient was calculated to verify the reliability of the scale used, and a correlation analysis of major variables was conducted. Third, a multiple additive moderation effect analysis was conducted using two moderating variables: economic well-being and level of education, in the relationship between parenting guilt and the mental health of single-parent women. At this time, multiple regression analysis of the simultaneous input method, including control variables, was used. In addition, to avoid multicollinearity among independent variables, mean-centering was performed [35], and Process Macro Model 2 was used. The moderating effect of the significance of the interaction term coefficient (XW) of the independent variable (X) and the moderating variable (W) was verified, and it is judged that there is a moderating effect if the confidence interval of the non-standardized estimate does not include zero at the confidence level of 95%. In addition, the regression coefficients of the 1 SD high group and the 1 SD low group were verified, focusing on the specific value of the moderating variable, that is, the average value. The research model set forth in this study is shown in Figure 1 below.

### 2.3. Measurement

#### 2.3.1. Mental Health

The General Health Questionnaire (GHQ-12; General Health Questionnaire-12), developed by Goldberg and Hillier [36] and adapted by Park et al. [37], was used to measure the mental health of single-parent women. It corresponds to a total of 12 questions, and each question is configured to respond to the 4-point Likert scale from never (1 point) to always (4 points). Positive questions (Nos. 1, 3, 4, 7, and 12) were used in reverse scoring, and the higher the score, the higher the mental health level of pain. Specific examples of questions include “feeling unhappy or depressed”, “under stress”, and “face up to problems”. The value of the reliability coefficient Cronbach’s α calculated in this study corresponds to 0.87. See Appendix A for the full scale. 

#### 2.3.2. Parenting Guilt

To gauge the parenting guilt of single-parent women, Mann and Thornberg [38] developed the Maternal Guilt Scale, which Jang [39] revised and expanded. Out of a total of 17 questions, 7 questions related to parental roles that can be applied to both employed and non-employed women were extracted and used. Each question consists of a five-point Likert scale ranging from strongly disagree (1 point) to strongly agree (5 points), and the higher the score, the higher the guilt of not playing a role as a parent. Examples of questions include “I feel that it is my fault when a child does something wrong”, “I feel guilty and regret after being angry with the child”, and “I am worried that the child will be emotionally anxious because I cannot give enough affection and love to the child”. The value of the reliability coefficient Cronbach’s α calculated in this study corresponds to 0.86. See Appendix B for the full scale. 

#### 2.3.3. Economic Well-Being

To assess the financial welfare of single-parent women, Lee [40] developed an economic well-being scale. In the original scale, a total of 13 questions were used by dividing them into three sub-factors: income adequacy, income improvement, and income stability, but in this analysis, only six key questions were used by excluding overlapping questions. Each question consists of a five-point Likert scale ranging from strongly disagree (1 point) to strongly agree (5 points), and the higher the score, the higher the subjective satisfaction with one’s economic level. Examples of questions include “I am satisfied with the current total asset level of the household”, “I am satisfied with the ability to prepare in case of an emergency”, and “I am satisfied with the savings rate to my household’s income”. The value of the reliability coefficient Cronbach’s α calculated in this study corresponds to 0.83. See Appendix C for the full scale.

#### 2.3.4. Level of Education

In order to measure the educational level of single-parent women, it was coded as 0 for those who graduated from high school or lower and 1 for those who graduated from 2-year college or higher. 

#### 2.3.5. Control Variable

In this study, an analysis was conducted by setting age, income, single-parent period, and the number of family members as control variables, which are demographic background variables judged to affect the mental health of single-parent women.

## 3. Results

Table 2 shows the descriptive statistics and correlation analysis results of the variables used in this analysis. First, the descriptive statistical results (average and standard deviation) of the main variables of this study (mental health, parenting guilt, economic well-being, and level of education) are as follows: In the case of mental health, the average of the questions was 2.98 points, which showed a high level of mental health pain. The range of the scale was 1~4 and the highest point was 4.0. That means that single-parent women “often” complain of depression, anxiety, and social dysfunction on average. The average of the questions for parenting guilt was 3.65 points, slightly higher than the median score on the scale (3 points). This shows that, on average, many women feel guilty about raising their children. The average score on the questionnaire for economic well-being was 1.71 points, which was lower than the median score (3 points) on the scale. This is the result of a large number of women showing low satisfaction with their economic situation. 

The average of the questions at the level of education was 0.4 points, which was lower than the median score (0.5 points) on the scale, indicating that there were many low-educated people. Next, before verifying the multiple additive moderating effects, Pearson’s coefficient of correlation was calculated to find out the correlation between the major variables. Parenting guilt showed a positive correlation with mental health (r = 0.58, *p* < 0.01), economic welfare showed a negative correlation with mental health (r = −0.35, *p* < 0.1) and parenting guilt (r = −0.14, *p* < 0.1).

Table 3 shows the results of multiple regression analysis to verify the moderating effect of economic well-being and academic level on the mental health of single-parent women. First, when the control variables are input, the higher the income level, the lower the level of mental health pain (coeff. = −0.08, *p* < 0.01). The independent variable, parenting guilt (coeff. = 0.36, *p* < 0.001) and the moderating variable, economic well-being (coeff. = −0.26, *p* < 0.001) were found to have a significant impact on mental health. In other words, the higher the parenting guilt and the lower the economic well-being, the higher the level of mental health pain. However, the level of education, a moderating variable, did not significantly affect the mental health of single-parent women (coeff. = 0.02, ns).

While the effect of the interaction term between parenting guilt and education level on mental health was statistically significant (coeff. = 0.16, *p* < 0.01), the interaction term between parenting guilt and economic welfare had no statistical significance (coeff. = −0.06, ns). This could also be confirmed by the confidence interval of non-standardized estimates at the 95% confidence level, and the interaction term between parenting guilt and education level had a moderating effect because the confidence interval did not include 0 at the 95% confidence level (LLCI = 0.05, ULCI = 0.26). The above results show that the effect of parenting guilt on mental health varies depending on the level of education. All variables used in the regression equation account for 43% of the total mental health variables (F = 34.59, *p* < 0.001). 

Changes in explanatory power due to the input of interaction terms are presented in Table 4. As the interaction term between parenting guilt and economic well-being was input, the change in explanatory power for mental health was not statistically significant (F = 1.65, ns), but as the interaction term between parenting guilt and education level was input, the explanatory power for mental health increased by 1% (F = 8.90, *p* < 0.01). In addition, as two interaction terms were input at the same time, the explanatory power for mental health increased by 1% (F = 5.25, *p* < 0.01). In order to examine this in more detail, the results of verifying the regression coefficients of the 1 SD high group and the 1 SD low group based on the average value are presented in Table 5. As shown in Table 5, when economic well-being and level of education were input together, the effect of parenting guilt on mental health in all groups was statistically significant. In other words, the results show that the greater the parenting guilt for all single-parent women, regardless of group, the higher the level of mental health pain. Among them, to compare the influence, the positive effect of parenting guilt on mental health was even greater in the group where economic welfare was below the average and women were highly educated (Effect = 0.49, *p* < 0.001).

## 4. Discussion

In this study, we examined how parenting guilt, economic well-being, and level of education affect mental health for single-parent women and verified how parenting guilt affects mental health depending on the two moderating variables (economic well-being and level of education). The following are the integration and discussions of meaningful results. First, the effects of economic well-being and parenting guilt on mental health, respectively, were consistent with several empirical studies. The higher the parenting guilt and the lower the economic well-being, the higher the level of mental health pain [14,16,41,42,43,44].

This raises the need to strengthen labor market policies and self-support programs to fulfill child support costs and overcome poverty for single parents. In addition, considering that economic help from family, relatives, and public institutions negatively affects the self-esteem of single parents [45], it is necessary to support the national effort to revitalize the self-support program or labor market rather than public assistance. Therefore, it is necessary to support household support services such as cleaning and housework so that single-parent women can have sufficient time and room for child rearing and to establish policies to help smooth the operation of counseling and mentoring services related to child rearing. Considering that only some local governments in Korea support domestic services individually, where national household service support policies are implemented, it is urgent to expand the scope of support and overhaul the legislation for nationwide support and management.

Second, as a result of verifying the multiple additive moderating effects, interesting results were found. The interaction term between parenting guilt and level of education had statistical significance, while the interaction term between parenting guilt and economic well-being did not produce significant results. Existing studies have shown that economic well-being affects parenting guilt [43], but in this study, results show that the effect of parenting guilt on mental health varies depending on the level of education. As a result of examining the size of the conditional effect of each group’s education level, it was found that the greater the guilt of parenting, the higher the level of mental health pain for women from all groups (a group of women with lower-than-average economic well-being and a low education level, a group of women with lower-than-average economic well-being and a high education level, a group of women with average economic well-being and a low education level, a group of women with average economic well-being and a high education level, a group of women with above-average economic well-being and a low education level, and a group of women with above-average economic well-being and a high education level) of single-parent families. These research results show the importance of education for single parents’ mental health. In particular, it suggests the importance of helping single parents develop and form self-esteem as human capital from a long-term perspective through higher education. In fact, in the case of single parents or single mothers, their studies are interrupted by pregnancy or childbirth, and the parenting circumstances following childbirth pose a barrier to single parents’ access to higher education [46]. In addition, the low level of education of single parents serves as a vulnerable factor in employment opportunities, so they do not even have access to economic activities, and low wages diminish life satisfaction, thereby lowering individuals’ self-esteem. Female single mothers, whether adolescents or undergraduates, must juggle their student status and parental responsibilities, causing them to experience emotions such as conflicting roles and remorse from their ideal parents [31]. 62% of the impoverished are high school dropouts, and 6% have bachelor’s degrees, whereas 72% of non-poor single mothers have bachelor’s degrees, according to Pandey et al. [47]. On the basis of the findings of this current study, it was anticipated that educational background would reduce the economic vulnerability of single parents, be a positive predictor of lifelong welfare, and serve as a positive expectation for children. 

Finally, the limitations of this study and suggestions for follow-up research are as follows: First, economic well-being and level of education were examined as moderating variables in this study, but it is believed that there may be more moderating variables that can affect the relationship between parenting guilt and mental health. Therefore, it is necessary to conduct an in-depth search for more variables, and accordingly, the scope of research methods needs to be expanded. For instance, stigma is increasingly recognized as being associated with poor health outcomes (including poor mental health), but these health disparities were not taken into account in this study. In subsequent studies, it will be necessary to consider the effect of stigma on negative mental health. Self-stigma is expected to reduce self-worth and hope for reaching goals, affecting parenting guilt and economic well-being, ultimately resulting in negative health conditions. If stigma considered, the future study would show the result effectively how parenting guilt, economic well-being, and level of education affect mental health for single-parent women. Second, this study targeted single-parent women, and it will need to be expanded to more diverse subjects. For example, research related to guilt about parenting has mainly focused on women, but it will need to be expanded to include research on men or grandparents from single-parent families. Third, this study had the advantage of randomly collecting single-parent families in Ulsan, but it cannot represent single-parent women across the country. Therefore, research will need to be conducted on more single-parent women. Regarding the problem of generalization, this study employed a great deal of description to contrast single-parent and two-parent families with relevant literature in order to make the distinction more apparent. Depending on the study’s recruitment strategy, this may not always be the correct description. In other words, the recruitment strategy for this study was to use a support group, which increases the likelihood of confronting single parents who are suffering as a result of their parenting role; this may bias the results. In addition, since bias in self-reported surveys cannot be ruled out, more objective data collection is needed to reflect the perspectives of family members of single-parent families or those related to welfare facilities. 

## 5. Conclusions

This study aimed to investigate the relationship between parenting guilt and mental health for 419 single-parent women. In addition, the additive moderating effect was verified to determine whether economic well-being and level of education were associated. As a result, the interaction term between parenting guilt and level of education was statistically significant, whereas the interaction term between parental guilt and economic well-being was not. These findings demonstrate education’s mental health benefits for single-parent women. It emphasizes helping them build self-esteem as human capital through higher education. A policy is required to assist single-parent women in improving their quality of life through education. Assisting single-parent women in acquiring childrearing knowledge and maintaining psychological health will also necessitate community involvement. For example, welfare centers or counseling centers should be able to locate them in order to impart a desire for higher education and alleviate the psychological strains of parenting. This study contributes in three ways. First, questionnaire-based quantitative data collection was conducted to objectively investigate the psychological recruitment of single-parent women. The majority of studies on single-parent women are qualitative or utilize limited information from single-parent survey data collected on a national scale (e.g., yes/no or categorical responses). This study allowed for a more empirical examination of single-parent women. Second, examining how the fundamental problems of single-parent families (e.g., parenting, economy, and education level) impact their mental health allowed for a deeper understanding of single-parent women. Third, by establishing two moderating variables to confirm multiple additive moderating effects, their dynamics could be investigated in greater detail.

## Figures and Tables

**Figure 1 healthcare-11-01814-f001:**
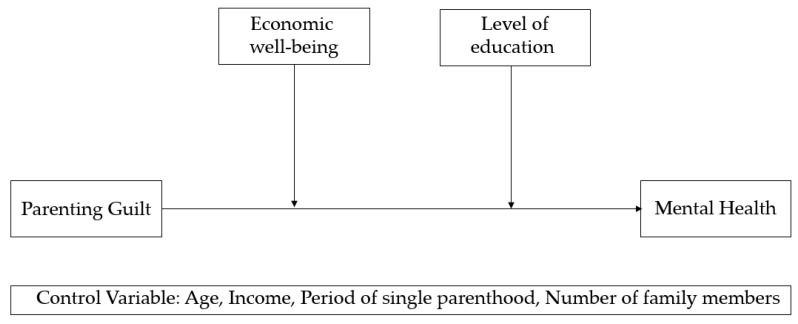
Research Model.

**Table 1 healthcare-11-01814-t001:** Demographic and sociological characteristics of participants (N = 419).

Variable	N	(%)	Variable	N	(%)
Age			Level of education		
20s	16	(3.85%)	elementary school	8	(1.9%)
30s	91	(21.7%)	middle school	29	(6.9%)
40s	230	(54.9%)	high school	214	(51.1%)
50s	79	(18.9%)	two-year college	92	(22.0%)
Over 60s	3	(0.7%)	four-year university	73	(17.4%)
Monthly income			graduate or higher	3	(0.7%)
no income	25	(6.0%)	Number of family members		
less than one million won	115	(27.4%)	2	161	(38.4%)
one million won~two million won	242	(57.8%)	3	164	(39.1%)
two million won~three million won	29	(6.9%)	4	67	(16.0%)
three million won~four million won	4	(1.0%)	5	25	(6.0%)
four million won~five million won	2	(0.5%)	6	2	(0.5%)
more five million won	2	(0.5%)	Residential area		
Period of single parenthood			Middle district	95	(22.7%)
less than 3 years	122	(29.1%)	Southern district	69	(16.5%)
more than 3 to less than 5 years	75	(17.9%)	Eastern district	58	(13.8%)
5 to less than 10 years	141	(33.7%)	Northern district	103	(24.6%)
more than 10 years	81	(19.3%)	Ulju county	94	(22.4%)

**Table 2 healthcare-11-01814-t002:** Descriptive statistics and correlation coefficients among variables.

	1	2	3	4	5	6	7	8
1. Mental health	1							
2. Parenting guilt	0.58 **	1						
3. Economic well-being	−0.35 **	−0.14 **	1					
4. Level of education	−0.02	−0.05	0.03	1				
5. Age	−0.06	−0.15 **	−0.06	0.01	1			
6. Income	0.06	0.03	−0.05	0.14 **	−0.01	1		
7. Period of single parenthood	−0.06	−0.04	−0.04	−0.03	0.28 **	−0.03	1	
8. Number of family members	0.03	0.06	−0.09	0.06	−0.07	0.18 **	−0.08	1
M (Mean)	2.98	3.65	1.71	0.40	3.91	2.15	2.43	2.91
SD (standard deviation)	0.56	0.79	0.59	0.49	0.76	1.41	1.10	0.91

** *p* < 0.01.

**Table 3 healthcare-11-01814-t003:** Multiple regression analysis: Multiple additive moderating effects and economic well-being and level of education between parenting guilt and mental health.

Variable	Coeff.	SE	t	LLCI	ULCI
(Constant number)	3.24	0.15	22.05 ***	2.95	3.52
Age	0.00	0.03	0.13	−0.05	0.06
Income	−0.08	0.03	−2.80 **	−0.13	−0.02
Period of single parenthood	−0.01	0.01	−1.05	−0.02	0.00
Number of family members	−0.01	0.02	−0.31	−0.06	0.04
Parenting guilt (A)	0.36	0.03	13.02 ***	0.31	0.42
Economic well-being (B)	−0.26	0.04	−7.12 ***	−0.33	−0.19
Level of education (C)	0.02	0.04	0.43	−0.07	0.10
A × B	−0.06	0.04	−1.29	−0.14	0.03
A × C	0.16	0.05	2.98 **	0.05	0.26
R	0.66				
R^2^	0.43				
F	34.59 ***				
*df* _1_	9				
*df* _2_	407				

** *p* < 0.01 *** *p* < 0.001.

**Table 4 healthcare-11-01814-t004:** Changes in R^2^ by Interaction terms.

Interactions	∆R^2^	F	*df* _1_	*df* _2_
Parenting guilt (A) × Economic well-being (B)	0.00	1.65	1	407
Parenting guilt (A) × Level of education (C)	0.01	8.90 **	1	407
A × B & A × C	0.01	5.25 **	2	407

** *p* < 0.01.

**Table 5 healthcare-11-01814-t005:** Regression coefficient of parenting guilt and mental health for low and high groups.

Economic Well-Being	Level of Education	Effect	SE	t	LLCI	ULCI
−1SD (−0.56)	Low (0)	0.33	0.05	7.17 ***	0.24	0.42
−1SD (−0.56)	High (1)	0.49	0.05	10.96 ***	0.40	0.58
M (0.03)	Low (0)	0.30	0.04	7.63 ***	0.22	0.37
M (0.03)	High (1)	0.46	0.04	12.33 ***	0.39	0.53
+1SD (0.62)	Low (0)	0.26	0.05	5.51 ***	0.17	0.36
+1SD (0.62)	High (1)	0.42	0.05	9.18 ***	0.33	0.51

*** *p* < 0.001.

## Data Availability

Not applicable. The participants in this study did not provide written consent for their data to be shared publicly; therefore, due to the sensitive nature of the research, the data is not available.

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
