# Peer review of "Influence of Parenting Guilt on the Mental Health among Single-Parent Women: Multiple Additive Moderating Effect of Economic Well-Being and Level of Education"

_healthcare, 2023, doi:10.3390/healthcare11131814_

Round 1

Reviewer 1 Report

There are some strengths to this paper, but also some major concerns. Firstly, the introduction highlights and critiques the social stigma experienced by single mothers, but the remainder of the paper appears to seek to reinforce this stigma. Stigma is increasingly recognised as associated poor health outcomes (including poor mental health), but the authors do not consider these health inequalities in their research.

The language throughout the paper generalises the experience of single mothers far too much, "Parents from single-parent families feel sorry for their children for being raised in a single-parent family compared to a two-parent family" - I don't think this is always true, and that your recruitment strategy was utilising a support group increases the likelihood of encountering single parents who are suffering due to their parenting role, this will skew results and should have been recorded in limitation. 

The authors refer to their child's "aggression" but this seems to come from nowhere- what childhood aggression are they referring to? There are a lot of assumptions made throughout this paper without the robust literature base to support the claims made (the literature exists, but should have been included). The literature is also sourced from countries all over the world, with variations on cultural expectations of motherhood which was not critiqued by the authors. 

Conclusion states questionnaires were used but methods describe 1:1 interviews and questionnaires, so clarity around this would be helpful.

This work is ambitious but results in some reductionist ideas and whilst the analysis is clear, and the paper itself is well constructed there is no evidence that this effectively demonstrates "how parenting guilt, economic well-being, and level of education affect mental health for women from single-parent families". Because stigma is not considered at any stage of this analysis and internal and external stigma have already been established within this paper as a fundamental challenge experienced by single mothers.

Proof reading required, grammatical and spelling errors throughout.

Author Response

Reviewer Report on Manuscript # healthcare-2411851 “Influence of Parenting Guilt on the Mental Health Among Women from Single-parent Families: Multiple Additive Moderating Effect of Economic Well-being and Level of Education”

Note: To facilitate our responses to all the comments, we have numbered each comment and provided the reviewer’s excerpts in italics. Our detailed responses follow in regular font with red color.

First of all, we sincerely appreciate the time and efforts you have provided. Your thoughtful suggestions have helped us substantially improve on our original submission. Please find our detailed response to your comments below.

Reviewer1.

  1. There are some strengths to this paper, but also some major concerns. Firstly, the introduction highlights and critiques the social stigma experienced by single mothers, but the remainder of the paper appears to seek to reinforce this stigma. Stigma is increasingly recognized as associated poor health outcomes (including poor mental health), but the authors do not consider these health inequalities in their research.

Thank you for making a valid point. According to your recommendation, we included your concerns as a limitation in the discussion section. Please see lines 371–375.

  1. The language throughout the paper generalizes the experience of single mothers far too much, "Parents from single-parent families feel sorry for their children for being raised in a single-parent family compared to a two-parent family" - I don't think this is always true, and that your recruitment strategy was utilizing a support group increases the likelihood of encountering single parents who are suffering due to their parenting role, this will skew results and should have been recorded in limitation.

Thank you for your thoughtful feedback. We revised the contents based on your suggestions. Please see lines 385–391.

  1. The authors refer to their child's "aggression" but this seems to come from nowhere- what childhood aggression are they referring to? There are a lot of assumptions made throughout this paper without the robust literature base to support the claims made (the literature exists, but should have been included). The literature is also sourced from countries all over the world, with variations on cultural expectations of motherhood which was not critiqued by the authors.

Thank you for recognizing an essential point. As you mentioned, we removed the section because it failed to account for the cultural diversity of the child's aggression. Please see lines 58–66.

4.Conclusion states questionnaires were used but methods describe 1:1 interviews and questionnaires, so clarity around this would be helpful.

Thank you for your comments. To clarify the sentence, we revised the writing. Please see lines 149 and 420.

  1. This work is ambitious but results in some reductionist ideas and whilst the analysis is clear, and the paper itself is well constructed there is no evidence that this effectively demonstrates "how parenting guilt, economic well-being, and level of education affect mental health for women from single-parent families". Because stigma is not considered at any stage of this analysis and internal and external stigma have already been established within this paper as a fundamental challenge experienced by single mothers.

We value your insightful suggestions. We rewrote this as a limitation in the study, considering your above suggestion (number 1). Please see lines 371–379.

Reviewer 2 Report

Introduction: Addresses important studies on the problem and the current state of the art. Clearly presents the objectives and research questions;

Material and methods: Perhaps it would be important to know the scales of Mental health; Parenting guilt; Economic well-being;

Results:The descriptive characteristics of the scales: Mental health; Parenting guilt; Economic well-being, should be more explicit.

Discussion: Confrontation of the main findings of this study with the existing bibliography, There is rigor in the description of limitations. Suggestions are displayed.

Conclusion: it presents the synthesis of the main results of the study, as well as the contributions to the improvement of knowledge about this reality and the proposal of measures that aim to improve this reality.

Author Response

Reviewer Report on Manuscript # healthcare-2411851 “Influence of Parenting Guilt on the Mental Health Among Women from Single-parent Families: Multiple Additive Moderating Effect of Economic Well-being and Level of Education”

Note: To facilitate our responses to all the comments, we have numbered each comment and provided the reviewer’s excerpts in italics. Our detailed responses follow in regular font with red color.

First of all, we sincerely appreciate the time and efforts you have provided. Your thoughtful suggestions have helped us substantially improve on our original submission. Please find our detailed response to your comments below.

Reviewer 2.

  1. Introduction: Addresses important studies on the problem and the current state of the art. Clearly presents the objectives and research questions;

We appreciate your comments. To improve the clarity of the sentence, we revised the text. Please see lines 125–140.

  1. Material and methods: Perhaps it would be important to know the scales of Mental health; Parenting guilt; Economic well-being;

We appreciate your suggestion. In accordance with your recommendation, we added the scales to the appendix. Please see lines 448-464.

  1. Results: The descriptive characteristics of the scales: Mental health; Parenting guilt; Economic well-being, should be more explicit.

Thank you for your suggestion. To clarify the sentence, we revised the writing. Please see lines 230-239.

  1. Discussion: Confrontation of the main findings of this study with the existing bibliography, There is rigor in the description of limitations. Suggestions are displayed.

We appreciate you bringing up a solid topic. In response to your suggestions, we updated the discussion section. Please see lines 298–394.  

  1. Conclusion: it presents the synthesis of the main results of the study, as well as the contributions to the improvement of knowledge about this reality and the proposal of measures that aim to improve this reality.

Thank you for your insightful comment. We have rewritten our conclusions entirely. Please see lines 396–420.    

Reviewer 3 Report

Authors A Kim and group examined the interactions between parenting guilt, economic well-being, and education level and their impact on the mental health of women from single-parent families. Based upon the quantitative analysis, they report a significantly important interaction term between parenting guilt and education level whereas failed to observe a significant interaction between parenting guilt and economic well-being.

While discussing their results, authors suggest towards an urgent need to expand the domestic support services through legislation, as well as support the national effort to revitalize the self-support program or labor market.

Authors have provided strong justification for their hypotheses and study design through a well written introduction, and presented adequate discussion and results comprising of relevant and necessary data tables.

Overall, the current manuscript is presented in a well-structured manner. Authors have cited appropriate and recent references through the manuscript. I do not have any major comments or revisions to suggest, however, I do have one suggestion. In the discussion, at lines 300-304, authors can briefly elaborate the observation of no significant interaction between parenting guilt and economic well-being as opposed to the findings of cited literature 24-28, 44.

Author Response

Reviewer Report on Manuscript # healthcare-2411851 “Influence of Parenting Guilt on the Mental Health Among Women from Single-parent Families: Multiple Additive Moderating Effect of Economic Well-being and Level of Education”

Note: To facilitate our responses to all the comments, we have numbered each comment and provided the reviewer’s excerpts in italics. Our detailed responses follow in regular font with red color.

First of all, we sincerely appreciate the time and efforts you have provided. Your thoughtful suggestions have helped us substantially improve on our original submission. Please find our detailed response to your comments below.

Reviewer 3.

Authors A Kim and group examined the interactions between parenting guilt, economic well-being, and education level and their impact on the mental health of women from single-parent families. Based upon the quantitative analysis, they report a significantly important interaction term between parenting guilt and education level whereas failed to observe a significant interaction between parenting guilt and economic well-being.

While discussing their results, authors suggest towards an urgent need to expand the domestic support services through legislation, as well as support the national effort to revitalize the self-support program or labor market.

Authors have provided strong justification for their hypotheses and study design through a well written introduction, and presented adequate discussion and results comprising of relevant and necessary data tables.

Overall, the current manuscript is presented in a well-structured manner. Authors have cited appropriate and recent references through the manuscript. I do not have any major comments or revisions to suggest, however, I do have one suggestion. In the discussion, at lines 300-304, authors can briefly elaborate the observation of no significant interaction between parenting guilt and economic well-being as opposed to the findings of cited literature 24-28, 44.

Thank you for discovering and assessing the worth of our paper. As suggested, we revised the corresponding matter. Please see lines 337-364.

Round 2

Reviewer 1 Report

Thank you to the authors for their significant work on this article.

In the introduction, please re-write this sentence for clarity "When the mother experienced a high work-interfering-with-family conflict and was different from the "ideal" mother she thought, she felt more guilty about her child"

All references to "single parent women" and "women in single parent families" should be changed to "single mothers" for consistency and clarity.

I am confused by your recommendations "Europe, where national household service support policies are implemented" - what is this reference to? I'm not aware of any such policies anywhere in Europe, but I may have misinterpreted what you mean. This also reinforces your expectations of 'ideal' mothering (spending more time with their children rather than home-making tasks) rather than responding to any of your research questions.

See previous comments

Author Response

(2nd Round)

Reviewer Report on Manuscript # healthcare-2411851 “Influence of Parenting Guilt on the Mental Health Among Women from Single-parent Families: Multiple Additive Moderating Effect of Economic Well-being and Level of Education”

Note: To facilitate our responses to all the comments, we have numbered each comment and provided the reviewer’s excerpts in italics. Our detailed responses follow in regular font with red color.

Again, we sincerely appreciate the time and efforts you have provided. Your thoughtful suggestions have helped us substantially improve on our original submission. Please find our detailed response to your comments below.

Reviewer1.

  1. In the introduction, please re-write this sentence for clarity "When the mother experienced a high work-interfering-with-family conflict and was different from the "ideal" mother she thought, she felt more guilty about her child"

 Thank you for your comments. To clarify the sentence, we revised the writing. Please see lines 68-70.

  1. All references to "single parent women" and "women in single parent families" should be changed to "single mothers" for consistency and clarity.

Thank you for your insightful comments. We modified all references to "single-parent women" to ensure consistency. Refer to lines 1-2, 12, 17-18, 26-27, 97, 139, 147, 149, 173, 189, 198, 210-211, 222-223, 229, 257, 264, 289-290, 300-301, 384, 389, 390-391, 402, 427, and 429-430.

  1. I am confused by your recommendations "Europe, where national household service support policies are implemented" - what is this reference to? I'm not aware of any such policies anywhere in Europe, but I may have misinterpreted what you mean. This also reinforces your expectations of 'ideal' mothering (spending more time with their children rather than home-making tasks) rather than responding to any of your research questions.

Thank you for making a valid point. We removed the sentence "unlike Europe" for clarity. Please see lines 332-333.
